# Perchlorate-Coupled Carbon Monoxide (CO) Oxidation by *Moorella glycerini*, an Obligately Anaerobic, Thermophilic, Nickel-Dependent Carboxydotroph

**DOI:** 10.3390/microorganisms11020462

**Published:** 2023-02-12

**Authors:** Marisa R. Myers, G. M. King

**Affiliations:** Department of Biological Sciences, Louisiana State University, Baton Rouge, LA 70803, USA

**Keywords:** carbon monoxide, perchlorate reduction, carbon monoxide dehydrogenase/acetyl CoA synthase, astrobiology

## Abstract

Many facultative and obligate anaerobes reduce perchlorate. Likewise, carbon monoxide (CO) oxidation has been documented in many aerobes, facultative anaerobes, and obligate anaerobes. A molybdenum-dependent CO dehydrogenase (Mo-CODH) and a nickel-dependent CO dehydrogenase (Ni-CODH) distinguish the former from the latter. Some Mo-dependent CO oxidizers (Mo-COX) couple CO oxidation to perchlorate reduction, but only at low concentrations of both under conditions that do not support growth in cultures. In contrast, CO-coupled perchlorate reduction has not been documented in Ni-dependent CO oxidizers (Ni-COX). To assess the potential for Ni-COX to reduce perchlorate, a model, obligately anaerobic homoacetogen, *Moorella glycerini* DSM 11254^T^, was cultivated with or without perchlorate, usiing CO or glycerol as its sole carbon and energy source. It grew with glycerol with or without perchlorate, and its maximum cell densities were only weakly affected by the perchlorate. However, when CO (at a 30% headspace concentration) was used as a carbon and energy source, perchlorate reduction supported greater cell densities and more rapid growth rates. The stoichiometry of CO uptake, perchlorate reduction, and chloride production were consistent with the cryptic pathway for perchlorate reduction with chlorite as an end product. Chloride production occurred abiologically in the medium due to a reaction between chlorite and the sulfide used as a reducing agent. These results provide the first demonstration of CO-coupled perchlorate reduction supporting growth in Ni-COX, and they provide constraints on the potential for perchlorate-coupled, anaerobic CO oxidation in engineered systems as well as terrestrial systems and hypothetical, sub-surface, serpentinite-hosted systems on Mars.

## 1. Introduction

Carbon monoxide (CO) participates in a wide range of processes, from cellular to global levels, and appears to have done so since the inception of life on Earth [1]. At present, CO oxidation occurs via two distinct mechanisms. A well-documented group of aerobes and facultative anaerobes use a Mo-dependent carbon monoxide dehydrogenase (Mo-CODH) with oxygen as a preferred terminal electron acceptor [1]. This group, referred to as Mo-dependent CO oxidizers (Mo-COX), plays a variety of important ecological roles, but is perhaps best known for its role in atmospheric CO consumption [1,2,3,4].

CO can also be oxidized anaerobically by Mo-COX. Early studies with non-extremophilic aquatic and soil isolates showed that nitrate served as a suitable electron acceptor for denitratation and denitrification when CO concentrations were ≤1000 ppm [5]. Later studies with moderate and extreme halophiles documented both nitrate and perchlorate coupling to CO at concentrations ≤ 100 ppm [6,7]. Isolates that reduced perchlorate did so using the “cryptic” pathway, which results in chlorite as an end product (Equation (1)), in contrast to the canonical pathway, which yields chloride and molecular oxygen [8]. These results are of interest due to the presence of CO in the atmosphere of Mars (approximately 800 ppm, [9,10,11]) and high concentrations of perchlorate in its regolith [12,13,14]. In particular, CO might support activity by extreme halophiles in hypothetical, near-surface brines when temperatures, water potentials, and chaotropicity are permissive [6,7,15,16].
2 CO + ClO_4_^−^ → 2 CO_2_ + ClO_2_^−^(1)

A second group of CO oxidizers, comprising strict anaerobes and referred to as Ni-dependent CO oxidizers (Ni-COX), uses Ni-containing CO dehydrogenases/acetyl CoA synthases (Ni-CODH) [17,18]. Genes for these enzymes appear to have been present in the last universal common ancestor (LUCA), where they could have contributed to a heterotrophic, mixotrophic, or autotrophic metabolism based on acetogenesis, methanogenesis, or hydrogenogenesis [19,20].

Although numerous metabolically diverse Ni-COX have been described [21], only a few have been reported to reduce perchlorate (e.g., [22,23,24,25]), and none of these have been shown to couple perchlorate reduction with CO oxidation. The hyperthermophilic, sulfate-reducing crenarchaeote, *Archaeoglobus fulgidus*, grows as a carboxydotroph by coupling CO oxidation to sulfate reduction or acetogenesis [26]. It also uses the cryptic pathway to reduce perchlorate to chlorite [24,27] but its ability to couple CO and perchlorate is unknown. Likewise, the thermophilic, homoacetogenic Firmicutes, *Moorella glycerini* and *M*. *sulfitireducens*, reduce perchlorate to chlorite and grow as carboxydotrophs using Ni-CODH [22,23,25,28], but their ability to couple perchlorate and CO has not been documented. 

The ability to beneficially couple CO oxidation and perchlorate reduction by obligate anaerobes has implications for microbial life on extraterrestrial worlds, including Mars. Specifically, relatively warm habitats with low-salinity water could occur at sub-surface sites on Mars with active serpentinization [29,30]. CO in such sites could support populations of Ni-COX capable of reducing perchlorate. Although perchlorate has not been documented in the Martian sub-surface, the presence of levels sufficient to support a CO-based metabolism cannot be dismissed.

To assess the possibilities and constraints for CO-coupled perchlorate reduction by Ni-COX, *Moorella glycerini* DSM 11254^T^ was used as a model cryptic perchlorate reducer that produces acetate during its growth on CO but that does not contain a chlorite-dismutase-like activity, unlike *M*. *perchloratireducens* An10 [22]. Results from this study indicated that CO oxidation supported perchlorate reduction, with a stoichiometry for CO oxidation, perchlorate reduction, and chloride formation consistent with chlorite production, followed by an abiological reduction to chloride at the expense of the sulfide present in the growth medium. In addition, perchlorate supported better growth with CO than CO alone, which suggested that perchlorate reduction was coupled beneficially to energy metabolism.

## 2. Materials and Methods

### 2.1. Isolate Cultivation

*Moorella glycerini* DSM 11254^T^ was obtained from the DSMZ GmbH (Braunschweig, Germany). It was cultivated in a modified version of DSMZ medium 793 (termed Moor1-S) with chloride concentrations reduced from 14.46 mM to 4.15 mM to facilitate chloride assays. Moor1-S contained the following (g L^−1^): KH_2_PO_4_, 0.33; (NH_4_)_2_SO_4_, 0.816; K_2_SO_4_, 0.772; MgSO_4_ × 7 H_2_O, 0.394; CaCl_2_ × 2 H_2_O, 0.33; yeast extract, 0.5; NaHCO_3_, 10; L-cysteine-HCl, 0.3; Na-resazurin solution (0.1% *w*/*v*), 0.5 mL; FeSO_4_ × 7 H_2_O (0.1% *w*/*v* in 0.1 N H_2_SO_4_), 2 mL; glycerol (87% stock solution, final medium concentration 41 mM), 3 mL; trace element solution SL-10 (DSMZ medium 320), 1 mL; selenite tungstate solution (DSMZ medium 385), 1 mL; and vitamin solution (DSMZ medium 141), 10 mL. The pH of the medium pH was adjusted to 6.5 using 1 M of sulfuric acid prior to filter sterilization. After sterilization, the medium was supplemented with filter-sterilized Na_2_S (heptahydrate, 5.0 g L^−1^). Serum bottles (60 mL or 160 mL) were sealed with blue butyl rubber stoppers containing 10 mL each of inoculated medium and an anoxic carbon dioxide headspace created by flushing with 100% CO_2_. The bottles were incubated with shaking (100 rpm) at 58 °C. Previous studies established that, under these conditions, perchlorate is not reduced and CO is not oxidized abiologically. Growth was monitored using liquid culture sub-samples obtained at intervals by needle and syringe for assays of absorbance at 600 nm (A_600_), performed using a Beckman DU-640 spectrophotometer (Beckman Instruments, Inc., Fullerton, CA, USA). The cell biomass was estimated from the absorbance values, as described by Weber and King [31].

### 2.2. Perchlorate-Coupled CO Oxidation

*M. glycerini* DSM 11254^T^ was assessed for its ability to couple perchlorate reduction to CO oxidation using inocula from freshly grown stationary phase cells, cultured in 160 mL serum bottles in Moor1-S with glycerol. Cells were harvested by centrifugation, washed in Moor1-S without glycerol, and resuspended in 50 mL of Moor1-S with or without glycerol, as required for subsequent treatments. A washed cell volume equivalent to 2.5% of the total experimental culture volume was used for inoculation. All sample headspaces were flushed with 100% CO_2_. Experimental treatments included triplicates of the following substrates in Moor1-S media: glycerol (41 mM); glycerol + 20 mM perchlorate; 30% CO (in Moor1-S without glycerol); and 30% CO + 20 mM perchlorate (in Moor1-S without glycerol). 

Headspace CO and O_2_ concentrations were assayed periodically by removing headspace sub-samples (0.4 mL) with a needle and syringe for gas chromatographic analysis. CO and oxygen concentrations were determined using an SRI 8610C gas chromatograph (SRI Instruments; Torrance, CA) equipped with a thermal conductivity detector and a 2-m × 0.32-cm (outer diameter) stainless steel column containing a molecular sieve 5A packing (80/100 mesh), operated with a helium carrier gas at 30 cm^3^ min^−1^ and an oven temperature of 60 °C. Perchlorate, chlorate, and chlorite concentrations were determined periodically using sub-samples (1.5 mL) obtained by needle and syringe. Perchlorate concentrations were measured with an ion-selective electrode, as described by Myers and King [6], with the exception that 100 µL of culture volume was used. Chlorate and chlorite concentrations were quantified via a colorimetric O-tolidine assay [32]. Chlorate assays were performed as described by Myers and King [6]; chlorite assays were performed identically to those for chlorate, with the exception that 4.8 M HCl was used and the absorbance was measured at 442 nm using a Beckman DU-640 spectrophotometer (Beckman Instruments; Fullerton, CA, USA). Chloride was determined using an altered version of the mercuric thiocyanate spectrophotometric method [33]. The mercuric thiocyanate assays contained a 1-mL final reaction volume mixed in the following order in a disposable cuvette: 38 µL sample, 346 µL water, 154 µL ferric ammonium sulfate solution, 77 µL mercuric thiocyanate solution, and 385 µL dioxane. These components were then mixed and allowed to incubate at room temperature for 10 min before the absorbance was read at 460 nm. A ferric ammonium sulfate solution was made fresh to avoid precipitation by mixing 6 g of ferric ammonium sulfate with 100 mL of 6 N nitric acid. A mercuric thiocyanate solution was prepared by mixing 300 mg of mercuric thiocyanate in 100 mL of 100% ethyl alcohol.

## 3. Results and Discussion

### 3.1. Growth and Perchlorate Reduction with Glycerol

*M. glycerini* DSM 11254^T^ grew in all conditions tested, though its growth varied with the treatment during a 27-d incubation (Table 1 and Appendix A). Lag times were somewhat shorter for cells grown with glycerol than for those grown with glycerol plus perchlorate (5.0 d vs. 6.8 d, respectively), but maximum growth rate constants for the two treatments were similar and did not differ statistically (Table 1: 0.454 ± 0.051 d^−1^ and 0.433 ± 0.079 d^−1^ for glycerol and glycerol plus perchlorate, respectively; *p* = 0.829), even though perchlorate was partially depleted in two replicates (59.5% and 53.5%) and used nearly completely in a third (96.3%). Likewise, the maximum cell densities did not differ statistically for glycerol incubations with or without perchlorate (Table 1: A_600_, 0.429 ± 0.033 and 0.475 ± 0.041 for glycerol and glycerol plus perchlorate, respectively; *p* = 0.435). These latter observations are similar to those of Balk et al. [22], who reported no difference in optical density for *M. glycerini* DSM 11254^T^ grown with fructose and with or without perchlorate, even though the perchlorate was reduced to a substantial degree in their study. This suggests that, although glycerol and fructose can be coupled to perchlorate reduction, the processes might not lead to a level of energy conservation sufficient to support appreciable growth.

Nonetheless, Balk et al. [22] showed that the cell densities of *M*. *perchloratireducens* An10 grown with fructose or methanol, or the cell densities of *M*. *mulderi* DSM 14980^T^ grown with fructose, increased in the presence of perchlorate. Reasons for the differences in growth among the isolates might be attributable to differences in energy conservation associated with the initial reduction step (perchlorate to chlorate). This possibility is supported by the fact that the cell densities were more than 2-fold higher for *M*. *mulderi* DSM 14980^T^ when grown with perchlorate, even though the amount of fructose used was the same and acetate production occurred at 50% of that for cells grown without perchlorate [22]. In contrast, the cell densities of *M*. *glycerini* DSM 11254^T^ were the same with and without perchlorate, fructose use decreased by 30%, and acetate was not produced at all during incubations with perchlorate.

Differences among isolates in the presence or activity of chlorite dismutase (cld; ClO_2_^−^ → Cl^−^ + O_2_) or a similar activity might also have had an impact. For example, *M*. *perchloratireducens* An10 expresses a cld-like activity [22] but does not harbor a canonical *cld* gene [25], and there is no evidence for such an activity or for a *cld* gene in *M. glycerini* DSM 11254^T^ [25]. Cld activity is a relevant factor because it could reduce the toxicity of the chlorite formed as an intermediate or end product of perchlorate reduction, thus enhancing growth.

### 3.2. Growth and Perchlorate Reduction with CO

CO also supported growth by *M*. *glycerini* DSM 11254^T^, consistent with previous reports and genomic evidence for the presence of Ni-CODH genes (23, see NCBI genome-accession number CP046224 and annotated Ni-CODH catalytic sub-unit at WP_156275103.1). However, the lag phase for growth on CO alone, 14.6 d, was considerably longer than for the growth with glycerol alone (5.0 d), and cell densities were also lower than those in the glycerol treatments (Table 1). Differences in lag times and cell densities were likely due to differences in the amount of carbon available for metabolism, a total of 2.14 mmol for CO and 6.15 mmol for glycerol, as well as lower free-energy yields for CO dissimilation (−151.5 kJ mol^−1^ for acetogenic glycerol fermentation versus −43.6 kJ mol^−1^ for acetogenic CO oxidation). However, in the presence of perchlorate, the lag time for growth with CO, 5 d, was considerably less than lag times for growth for CO alone or for growth with glycerol plus perchlorate (Table 1). In addition, the maximum cell densities for cultures grown with CO plus perchlorate substantially exceeded those for cultures grown with CO only (Figure 1), although the total amount of cell biomass produced during growth with perchlorate, 1.8 ± 0.2 mg (estimated from A_600_), amounted to only 4.3% of the amount of CO used, assuming a 50% carbon content for cell mass. These cell yields were consistent with the Ni-CODH-dependent anaerobic growth reported for Ni-COX and were higher than for cells grown without perchlorate. The results suggest that CO was beneficially coupled to perchlorate reduction, implying an energy conservation that supported enhanced growth.

The reasons for the different growth outcomes for the perchlorate reduction coupled to glycerol versus CO are unclear. A possible explanation can be inferred from the fact that the extent of perchlorate reduction varied among the three replicates incubated with glycerol. In the replicate with nearly complete perchlorate uptake, cell densities were distinctly higher than for the cells incubated with glycerol alone; they were also higher than the cell densities for the two replicates with a partial perchlorate uptake. Thus, variability in perchlorate reduction, perhaps involving variability in the extent of homoacetogenic metabolism, could lead to different growth yields.

Interestingly, chloride did not accumulate in the replicates with partial perchlorate depletion, which might indicate that chlorate was the end-product. In contrast, chloride was produced stoichiometrically in the replicate with substantial perchlorate reduction, which can be attributed to chlorite formation, followed by abiological reduction to chloride at the expense of the sulfide in the growth medium [34]. Nonetheless, since genes for chlorite dismutase have not been documented for *M*. *glycerini* DSM 11254^T^, mechanisms involved in perchlorate reduction and the controls of its use and variability remain uncertain. 

### 3.3. CO-Coupled Chloride Production

Chloride was also produced by all replicates in incubations with CO and perchlorate, with the decreases in perchlorate paralleled by increases in chloride (Figure 2). Over the course of the incubations, perchlorate declined by a total of 600 ± 2 µmol, while chloride increased by a total of 555 ± 19 µmol. The ratio of perchlorate consumed to chloride produced, 0.92 ± 0.03, was not significantly different than a test value of 1 (Student’s *t*-test, *p* = 0.147), nor were the amounts of perchlorate consumed and chloride produced significantly different (Student’s *t*-test, *p* = 0.077). Nonetheless, the difference between the perchlorate reduced and chloride formed is consistent with the formation of a small amount of chlorate −45 µmol or less—which might have been formed during cryptic perchlorate reduction. Under the incubation conditions used in the study, it would not have been reduced abiologically, and thus would not have contributed to chloride formation. However, we were unable to detect either chlorate or chlorite, indicating that the former did not accumulate as a free end product while the latter was consumed by sulfide.

The total CO consumption during incubations with perchlorate, 1761 ± 12 µmol, exceeded that for CO alone, 1321 ± 33 µmol (Students *t*-test, *p* = 0.021), which likely accounts for the differences in the maximum cell density for the two treatments. Using a 2:1 stoichiometry for the CO-coupled perchlorate reduction to chlorite (Equation (1)), the observed level of chloride production implies a total uptake of 1110 µmol CO, with an additional 45 µmol of CO uptake inferred if the difference between chloride production and perchlorate reduction can be attributed to chlorate formation (a 1:1 stoichiometry). The difference between the estimated and observed CO uptake (1761 µmol − 1155 µmol = 606 µmol) likely represents the CO incorporated into biomass and acetate, a known end product of homoacetogenic metabolism by *M*. *glycerini* DSM 11254^T^ [25,28]. Although acetate was not measured in this study, levels of approximately 8 mM can be inferred from the amount of CO not accounted for by chloride and biomass production. This implies concurrent perchlorate reduction and acetogenesis, which has been observed for *M*. *perchloratireducens* An10 and *M*. *mulderi* DSM 14980^T^ during the dissimilation of fructose but has not been reported for *M*. *glycerini* DSM 11254^T^ [22].

## 4. Summary and Conclusions

The results of this study provide the first evidence for the growth and CO-coupled perchlorate reduction by Ni-COX using the cryptic pathway, the product of which, chlorite, was reduced approximately stoichiometrically to chloride by sulfide. In contrast to previous reports for Mo-COX (Myers and King, 2018), which could only reduce perchlorate with CO at concentrations < 100–1000 ppm, *M*. *glycerini* DSM 11254^T^ used CO at concentrations of 30%. This suggests that Ni-COX are less subject to CO inhibition of perchlorate reduction than Mo-COX. These observations might aid in the design of systems for in situ or ex situ perchlorate bioremediation using CO as an inexpensive electron donor [35], although additional effort is necessary to determine how *M*. *glycerini* responds to ambient perchlorate concentrations, which are much lower than those used in this study. However, it might be equally or more interesting to assess the capabilities of *M*. *perchloratireducens* An10, which has been reported to reduce perchlorate and express a chlorite-dismutase-like activity [22]. Although *M*. *perchloratireducens* An10 has not yet been reported to oxidize CO, inspection of its genome sequence reveals the presence of a canonical Ni-CODH [36] which could potentially support CO-coupled perchlorate reduction as described here but without the requirement for sulfide as a chlorite sink. Such a capacity would represent a new dimension for perchlorate dissimilation with implications for Mars, where perchlorate is exceptionally abundant in the regolith and CO occurs at relatively high atmospheric concentrations. Although extreme conditions at the surface might preclude microbial activity, the sub-surface, especially in regions that support serpentinization, could be conducive to extant life. Transport of perchlorate to such systems from a surface reservoir could support perchlorate reduction by Ni-COX anaerobes with traits similar to those of *M*. *glycerini* DSM 11254^T^ and possibly *M*. *perchloratireducens* An10.

## Figures and Tables

**Figure 1 microorganisms-11-00462-f001:**
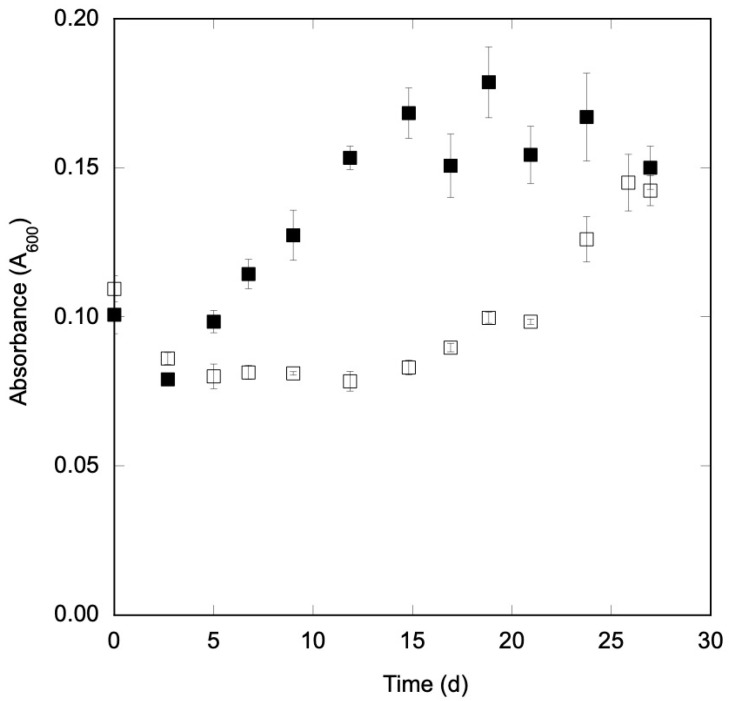
Growth (A_600_) of *Moorella glycerini* DSM 11254^T^ with a headspace containing 30% carbon monoxide with (solid squares) or without (open squares) 20 mM perchlorate. Values represent means of triplicates ± 1 standard error.

**Figure 2 microorganisms-11-00462-f002:**
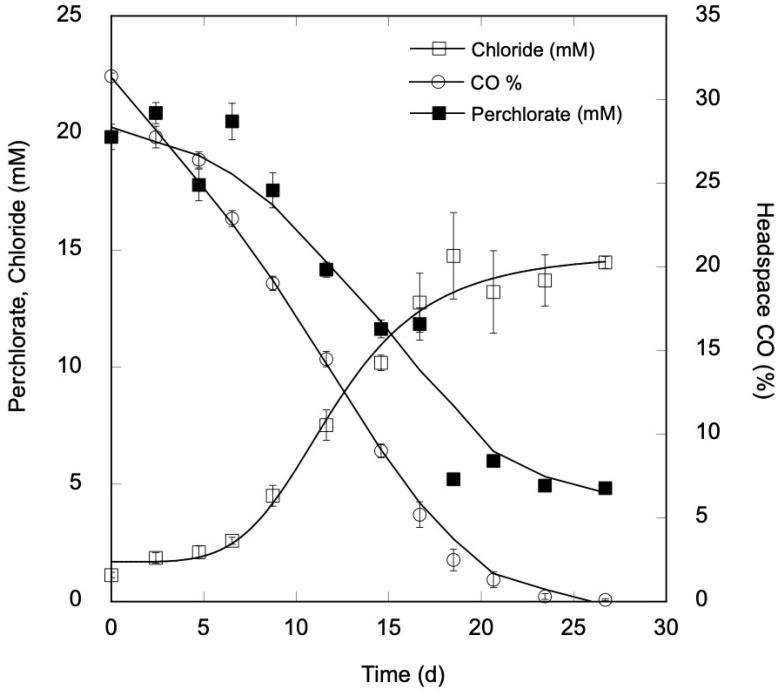
Uptake of CO (open circles) perchlorate (closed circles) and chloride production (open squares) by *Moorella glycerini* DSM 11254^T^ incubated with 20 mM perchlorate and a headspace containing 30% carbon monoxide. Data are means of triplicate determinations with ±1 standard error.

**Table 1 microorganisms-11-00462-t001:** Maximum growth rate constants (k_max_, d^−1^), maximum cell densities (A_600_ max), and lag phases (d) for *M*. *glycerini* DSM 11254^T^ incubated under various conditions as indicated. Values are means of triplicate determinations ± 1 standard error for k_max_ and A_600_ max; lag phases are means of triplicates that were the same for each replicate.

Treatment	k_max_ (d^−1^)	A_600_ max	Lag Phase (d)
Glycerol only	0.454 ± 0.051	0.429 ± 0.033	5.0
Glycerol + 20 mM perchlorate	0.433 ± 0.079	0.475 ± 0.041	6.8
CO only	0.093 ± 0.019	0.154 ± 0.001	14.6
CO + 20 mM perchlorate	0.155 ± 0.030	0.187 ± 0.012	5.0

## Data Availability

Data are available from the Appendix A and the authors.

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
