# Peer review of "Perchlorate-Coupled Carbon Monoxide (CO) Oxidation by Moorella glycerini, an Obligately Anaerobic, Thermophilic, Nickel-Dependent Carboxydotroph"

_microorganisms, 2023, doi:10.3390/microorganisms11020462_

Round 1

Reviewer 1 Report

To assess the potential for Ni-COX to reduce perchlorate, the authors cultivated Moorella glycerini DSM 11254T with/without perchlorate with CO or glycerol as sole carbon and energy sources. It grew with glycerol with or without perchlorate but maximum cell densities were only weakly affected by perchlorate. However, when CO (30% headspace concentration) was used as a carbon and energy source, perchlorate reduction supported greater cell densities and more rapid growth rates. The stoichiometry of CO uptake, perchlorate reduction, and chloride production were consistent with the cryptic pathway for perchlorate reduction with chlorite as an end-product. Chloride production occurred abiologically in the medium due to a reaction between chlorite and sulfide used as a reducing agent.

The authors conclude that these results provide the first demonstration of CO-coupled perchlorate reduction supporting growth in Ni-CODH. They also discuss the potential for perchlorate-coupled anaerobic CO oxidation in various systems (“engineered systems as well as terrestrial systems and hypothetical sub-surface serpentinite-hosted systems on Mars”). I have several concerns that need to be addressed before publication.

1. What limits the cell yield and the growth rate on CO relative to glycerol. They suggest: “differences in the amount of carbon available for metabolism, a total of 2.14 mmol 182 for CO and 6.15 mmol for glycerol, as well as lower free energy yields for CO dissimilation.” The manuscript should explicitly include these free energy yields for CO dissimilation to support their statement.  

2. This is confusing: “Although numerous metabolically diverse Ni-COX have been described (21), only a few have been reported to reduce perchlorate (e.g., 22-25) and none of these have been shown to couple perchlorate reduction to CO oxidation.” In looking at the papers, I think they mean that acetogens containing Ni-CODH have been reported to reduce perchlorate, but no one has shown that the Ni-CODH can couple perchlorate reduction to CO oxidation? Otherwise, the statement seems contradictory.

3. Line 179: “… genomic evidence for the presence of Ni-CODH genes.” Please provide a reference for this statement.

4. Line 192: “and imply energy conservation that supported enhanced growth.” Can you suggest how much ATP would have been generated by the addition of perchlorate?

5. Why do the authors think that there are such long lags with growth on CO in the absence of perchlorate relative to its presence?

6. The authors refer to a lot of variability in growth – is this variability accounted for in the statistics shown in Table/Fig. 1 1? The error bars are very small in Fig. 1; if these are the replicates they are referring to, I don’t think this is significant variability.

7. The authors should add the growth curves for the glycerol +/- perchlorate in Fig. 1. If so they should use different scales on right and left axes so that the data for the CO-grown cells are clearly discerned.

8. “Nonetheless, a small amount of chlorate -- 45 μmol or less -- might have been formed during cryptic perchlorate reduction.” The authors included a chlorate assay in the Methods section; why wasn’t the data presented? Perhaps chlorate was an intermediate in the production of chloride. On the other hand, the data for perchlorate loss mirrors that for Chloride formation, suggesting this is not an intermediate. Did the authors fit the data for perchlorate loss to include a lag as they did for chloride formation. The CO data do not show an obvious lag; could this be related to the redox stoichiometry, with 4 electrons required to reduce perchlorate, but only two involved in CO oxidation.

9. Can Moorella glycerini grow with chlorate as electron acceptor?  

10. “… approximately 8 mM [acetate] can be inferred.” I don’t understand why they didn’t assay acetate production. This seems like an oversight, when studying the growth of an acetogen. They also state: “This implies concurrent perchlorate reduction and acetogenesis…”. Why leave this at the level of an implication?

Author Response

Thank you for the helpful comments. We are sorry for the delay. Emails from the editor were gobbled up by Gmail and then some travel got in the way. We have copied the questions/comments and then added replies in italics.

  1. What limits the cell yield and the growth rate on CO relative to glycerol. They suggest: “differences in the amount of carbon available for metabolism, a total of 2.14 mmol 182 for CO and 6.15 mmol for glycerol, as well as lower free energy yields for CO dissimilation.” The manuscript should explicitly include these free energy yields for CO dissimilation to support their statement.  

We have added the relevant values to the text as suggested, -151.5 kJ mol-1 for acetogenic glycerol fermentation versus -43.6 kJ mol-1 for acetogenic CO oxidation, l. 183-184

  1. This is confusing: “Although numerous metabolically diverse Ni-COX have been described (21), only a few have been reported to reduce perchlorate (e.g., 22-25) and none of these have been shown to couple perchlorate reduction to CO oxidation.” In looking at the papers, I think they mean that acetogens containing Ni-CODH have been reported to reduce perchlorate, but no one has shown that the Ni-CODH can couple perchlorate reduction to CO oxidation? Otherwise, the statement seems contradictory.

This is the correct interpretation. Given that, we have left the original text.

  1. Line 179: “… genomic evidence for the presence of Ni-CODH genes.” Please provide a reference for this statement.

References added as requested; however, the M. glycerini genome annotation has not yet been published; it is instead available through NCBI with accession number CP046244; we have added this to the text along with the accession number for the CODH catalytic sub-unit, WP_156275103.1. This should be sufficient for readers to track down the Ni-CODH genes and evaluate the basis for claims about what they are.

  1. Line 192: “and imply energy conservation that supported enhanced growth.” Can you suggest how much ATP would have been generated by the addition of perchlorate?

This is challenging. The reaction of 2 CO + perchlorate  2 CO2 + ClO2- has a DG of about -580 kJ per reaction. This is more than ample for ATP synthesis. However, synthesis of ATP at the levels implied by the DG would require an ETS that has not been fully documented in Moorella. We also cannot say whether energy conservation occurs at both perchlorate and chlorate reduction steps. Thus, we are reluctant to guesstimate ATP yields is beyond the scope of the manuscript.

  1. Why do the authors think that there are such long lags with growth on CO in the absence of perchlorate relative to its presence?

This is an interesting question. One could speculate that this reflects some additional energy conservation in the presence of perchlorate. However, that takes us back to point 4 so all we can do in that context is guess. However, a similar phenomenon was observed by Balk et al. (2008) in their study of ‘Moorella perchloratireducens’ grown with methanol with or without perchlorate. They likewise reported increased cell yields when the isolate was grown with fructose plus perchlorate relative to growth without perchlorate. Thus, even though the mechanism isn’t clear, the data overall strongly imply that perchlorate reduction can be coupled beneficially to energy conservation that in turn supports growth.

  1. The authors refer to a lot of variability in growth – is this variability accounted for in the statistics shown in Table/Fig. 1 1? The error bars are very small in Fig. 1; if these are the replicates they are referring to, I don’t think this is significant variability.

Our comments about variability were constrained to a discussion of the results for incubations of glycerol with perchlorate. In this case, we noted that there was variability among the replicates in growth, perchlorate reduction and chloride production. We are uncertain to what the observed variability might be attributed but note that is was distinct from the much more consistent results with CO. However, since the focus of the study was the CO/perchlorate coupling, we did not pursue the glycerol results further. The primary reason for using glycerol at all was so that we would have a metabolic “control,” since M. glycerini was original isolated and described as a glycerol fermentor. We are also unable to compare results in this study with those of Balk et al. (2008), who used fructose and perchlorate, since they used duplicates and reported only means without a range.

  1. The authors should add the growth curves for the glycerol +/- perchlorate in Fig. 1. If so they should use different scales on right and left axes so that the data for the CO-grown cells are clearly discerned.

We have included a supplemental table with these data. Again, this wasn’t the focus of the study, so we have retain the original F1. However, those who are interested will have the data for both glycerol and CO.

  1. “Nonetheless, a small amount of chlorate -- 45 μmol or less -- might have been formed during cryptic perchlorate reduction.” The authors included a chlorate assay in the Methods section; why wasn’t the data presented? Perhaps chlorate was an intermediate in the production of chloride. On the other hand, the data for perchlorate loss mirrors that for Chloride formation, suggesting this is not an intermediate. Did the authors fit the data for perchlorate loss to include a lag as they did for chloride formation. The CO data do not show an obvious lag; could this be related to the redox stoichiometry, with 4 electrons required to reduce perchlorate, but only two involved in CO oxidation.

We did not detect chlorate or chlorite or chlorate accumulation in the medium (line 244-246), so it is possible that neither were produced transiently. Our stoichiometry does not include them in any case. Instead, our comments simply reflect the possibility that is implied by the difference between what was observed and what might be expected theoretically. We modified the text to emphasize this point.

  1. Can Moorella glycerini grow with chlorate as electron acceptor?

We are not aware of any separate evidence for growth with chlorate alone and did not explore that possibility, since our focus was on perchlorate, which is of greater interest in environmental and astrobiological contexts.

  1. “… approximately 8 mM [acetate] can be inferred.” I don’t understand why they didn’t assay acetate production. This seems like an oversight, when studying the growth of an acetogen. They also state: “This implies concurrent perchlorate reduction and acetogenesis…”. Why leave this at the level of an implication?

With hindsight, it would have been great to assay acetate, although it is not part of our routine suite of analytes. The possibility that acetate was produced emerged after analyzing the results and reviewing substrate stoichiometries, at which point conducting additional assays was no longer feasible. We also note that the focused goal of the work was to assess the potential for CO to serve as an electron donor for perchlorate reduction rather than following the fate of the CO-carbon per se.

Reviewer 2 Report

In this paper, the perchlorate-coupled carbon monoxide (CO) oxidation by Moorella glycerini was investigated. The authors have published lot of related paper in this field. Although they think that this paper provided the first evidence for growth and CO-coupled perchlorate reduction by Ni-COX using the cryptic pathway. But after I review the experiment design and the result, I think the author did not provide the enough evidences to support this opinion.

1. All the experiments and results are too few, which can not meet the level of this journal.

2. Is the M. glycerini DSM 11254T can express Ni-CODH? Or how do you know the key emzyme is Ni-CODH and not Mo-CODH? In this paper, Please provide ample evidences to support this view. Only genomic evidence for the presence of Ni-CODH genes from other papers is not enough.

3. The mechanisms for perchlorate-coupled carbon monoxide (CO) oxidation by M. glycerini should be cleared.

Author Response

Thank you for the helpful comments. We are sorry for the delay. This resulted from emails that disappeared and travel. Comments are copied below with responses in italics.

  1. All the experiments and results are too few, which can not meet the level of this journal.

We truly appreciate the reviewer’s perspective but at the same time we respectfully argue that the results we present establish what we set out to do, which is to demonstrate whether or not CO could be coupled to perchlorate reduction. We observed perchlorate reduction and CO uptake along with cell growth in a context that has no other plausible explanation. While we agree that a more detailed analysis could address many interesting questions, e.g., concentration dependencies, temperature optima, expression profiles and so on, the study as designed allows us to make the limited conclusions in the manuscript.

  1. Is theM. glyceriniDSM 11254T can express Ni-CODH? Or how do you know the key emzyme is Ni-CODH and not Mo-CODH? In this paper, Please provide ample evidences to support this view. Only genomic evidence for the presence of Ni-CODH genes from other papers is not enough.

Again, we appreciate the reviewer’s perspective but suggest that this study is no different than hundreds, if not thousands, of other studies that draw on published genomic or enzymatic evidence to draw conclusions about outcomes in a separate study. In this case, there is no evidence for the Mo-CODH genes in M. glycerini DSM 11254T or any other Moorella isolate, nor is there any evidence that both forms of CODH co-occur in any other taxa. All the available evidence tells us that CO oxidizers have one or the other form of CODH, not both.

Genomic evidence tells us that M. glycerini DSM 11254T possess the genes for Ni-CODH but even in the absence of such evidence, we know that M. glycerini DSM 11254T is a homoacetogen and that homoacetogens use the Wood-Ljungdahl pathway for acetate formation and this requires the Ni-CODH (and genes) that support CO uptake.

We respectfully suggest that the overwhelming weight of available evidence provides only one explanation that meets the “Occam razor test” of a parsimonious conclusion: previous studies have documented CO uptake by a variety of Moorella species (Jiang et al. 2009, Arch Microbiol 191:123-131) and M. glycerini DSM 11254T specifically (Alves et al. 2013, citation in text); these taxa all contain genes for Ni-CODH but not Mo-CODH (see Redl et al. 2020 for genome analyses; citation in text); Ni-CODH has long been known as the key enzyme for CO uptake in obligate anaerobes.

To make that point, we have provided the accession number for the M. glycerini DSM 11254T genome along with the accession number for its Ni-CODH.

  1. The mechanisms for perchlorate-coupled carbon monoxide (CO) oxidation byM. glycerini should be cleared.

To our knowledge, there is only one mechanism, which in general has to involve the oxidation of CO by Ni-CODH, the details of which have long been documents. Reducing equivalents from the Ni-CODH must be transferred to a perchlorate/nitrate reductase, but in this case the details are less well known. For the purposes of the manuscript at hand, we feel we have covered the basics sufficiently.

Reviewer 3 Report

The manuscript fits well to the scientific profile of “Microorganisms” and could be published after minor revision. The following points should be addressed before the final acceptance of the paper:

 L47 – reference in numerical format

L56 - ) bracket

L86 – Germany

L95 – “medium was supplemented with filter sterilized Na2S (heptahydrate, 5.0 g L-1)”. Such a high concentration of Na2S looks very strange at first glance. Usually, the Na2S concentration used as a reducing agent is ten times lower. Did you increase Na2S to create more favorable conditions for abiotic chlorite reduction? Please, note it in the text.

L102 – Specify the coefficient between absorbance and cell numbers of Moorella glycerini.

L142 - A 27 day incubation is quite a long time. Why did Moorella glycerini grow so slowly? The doubling time of this strain on glycerol is about 4-5 hours. Can a high concentration of Na2S cause growth inhibition? It would be great if you could discuss this issue.

L146, 173, 219, 221, 224, 234 – should it be “replicates” or “replicas”?

L166 – reference in numerical format

L192 – Provide the results on growth of Moorella glycerini without CO and glycerol on a medium with 500 mg/l of yeast extract.

L242 – Did you detect any chlorite accumulation experimentally? Please, indicate.

L276 - ) bracket

Author Response

Thank you for the helpful comments. We are sorry for the delay in responding. The editor's emails were gobbled up by gmail and we did not see them for a couple of weeks. The original points are copied below with responses in italics.

L47 – reference in numerical format

corrected

L56 - ) bracket

corrected

L86 – Germany

corrected

L95 – “medium was supplemented with filter sterilized Na2S (heptahydrate, 5.0 g L-1)”. Such a high concentration of Na2S looks very strange at first glance. Usually, the Na2S concentration used as a reducing agent is ten times lower. Did you increase Na2S to create more favorable conditions for abiotic chlorite reduction? Please, note it in the text.

corrected

L102 – Specify the coefficient between absorbance and cell numbers of Moorella glycerini.

We used the following relationship to estimate biomass from OD:

µg dry weight ml-1 = (364.74)(Abs600 nm) + (6.7)(Abs600 nm)

L142 - A 27 day incubation is quite a long time. Why did Moorella glycerini grow so slowly? The doubling time of this strain on glycerol is about 4-5 hours. Can a high concentration of Na2S cause growth inhibition? It would be great if you could discuss this issue.

Growth on CO with perchlorate was actually reasonable and consistent with results from Balk et al. (2008); growth on CO without perchlorate was slower but this could be due to lower energy yields. In addition, Balk et al. (2008) observed a lag of about 6 days for growth of their isolate on methanol alone but a lag of about 3 days for growth on methanol plus perchlorate; yields were also higher with perchlorate. Thus, our growth in our study might be a bit slower but it wasn’t remarkably so. We also not that the extended incubation allowed us to ensure that we could observe maximum perchlorate uptake and chloride production. We comment on this from l. 142-l. 168

L146, 173, 219, 221, 224, 234 – should it be “replicates” or “replicas”?

They should all be replicates.

L166 – reference in numerical format

corrected

L192 – Provide the results on growth of Moorella glycerini without CO and glycerol on a medium with 500 mg/l of yeast extract.

We have added a supplemental table with these and other results.

L242 – Did you detect any chlorite accumulation experimentally? Please, indicate.

No, we did not. We have included this is the text (l. 244-246).

L276 - ) bracket

corrected

Round 2

Reviewer 2 Report

The manuscript has been sufficiently improved to warrant publication in Microorganisms.

Author Response

thanks